# Do Migration and Acculturation Impact Somatization? A Scoping Review

**DOI:** 10.3390/ijerph192316011

**Published:** 2022-11-30

**Authors:** Antonello Barbati, Alessandro Geraci, Fabiana Niro, Letizia Pezzi, Marco Sarchiapone

**Affiliations:** 1Department of Medicine and Health Science, University of Molise, 86100 Campobasso, Italy; 2Ospedale Ca Foncello, 31100 Treviso, Italy; 3Department of Pathology, Federico II University of Naples, 80138 Naples, Italy; 4Rehabilitation Unit, ASST Cremona-Ospedale di Cremona, 26100 Cremona, Italy

**Keywords:** somatization, migration, acculturation

## Abstract

Somatization is a phenomenon in which the individual experiences physical symptoms attributable to mental projections. It is a widely used term in common parlance to figuratively describe a stressful situation. Syndromes directly related to the mind have been described; pathologies are influenced by somatization. However, the extent of somatization is also related to social and cultural factors. In fact, each culture expresses varying levels of somatization characteristic of the country of origin. A disease can even manifest with different symptoms in different ethnic groups. The migration process arises from the need for change on the part of those who undertake it and culminates in the integration of the person in the host country. This process induces changes in the person of a psychological nature, which also affects somatization. In fact, the most integrated subjects show levels of somatization comparable to those of the host country. These considerations support the thesis that psychological changes are an integral part of health and can affect the development of organic and somatized pathologies.

## 1. Introduction

Somatization is responsible for the influence of the mind on the human body. Although the concept of somatization is associated with concern for physical symptoms, it is widely demonstrated that various disorders of a psychological nature significantly affect the clinical history of numerous pathologies. The complexity of these relationships has led to the field of psycho-neuro-endocrine-immunology (PNEI), which studies the interactions between the central nervous, endocrine, and immune systems and their effect on human and animal behavior. The effects of various mental illnesses (post-traumatic stress disorders, affective and eating disorders, etc.) are also responsible for altering interoceptive processes, or perception of internal information, such as breathing, gastrointestinal peristalsis, and sense of hunger, which can lead to sensitization phenomena [1].

Somatization, therefore, represents a modality that the mind uses to communicate with the body. In other words, the externalization of mental suffering is manifested in unique and multiple ways [2]. 

Several mental pathologies closely related to this phenomenon have been described, such as depression, anxiety, and post-traumatic stress disorder [3,4,5]. This mechanism is individual but is affected by cultural and social habits. Normally, somatization levels are higher in women with low socioeconomic levels and education [6,7,8]. However, the extent of somatization is also associated with cultural phenomena. Cultural differences play an important role in somatization since it is expressed and experienced socio-culturally [9]. For example, Puerto Ricans define particularly stressful events as “ataque de nervios” (attack of nerves) with both physical and behavioral symptoms typically displayed [10]. In addition, the sociomoral situation, personality, and language influence how a disease is labeled [11]. 

In Western societies, somatic distress is subordinate to the mind and is considered a manifestation of psychological distress [12]. In other cultures, however, this approach is very different. 

For example, among non-Western cultures, the Chinese and East Indians manifest somatic symptoms more frequently and are often associated with depressive pictures [13]. In these populations, this modality is frequently used to communicate psychological distress [14]. This is not surprising considering how widespread Hindu, Buddhist, and Confucian traditions are in these cultures, where psychological and somatic distress are synonymous concepts [12]. In China, until the 1980s, psychology was given little consideration, and most psychiatric patients were defined as neurasthenic. Patients were diagnosed with “shenjing shuairuo”, or neurological weakness, which includes a series of somatic symptoms such as insomnia, fatigue, and dizziness along with cognitive symptoms such as poor memory or unpleasant thoughts and emotional symptoms such as irritability, excitability, or nervousness, as well as depressive-type symptoms [15]. For this reason, a social support system called “guanxi” is well established from a young age. This system comprises a complex social and economic relationship that represents a useful point of reference and a rescue net in case of need [16]. 

Other countries, such as Japan, provide similar social support in traditional culture, typically consisting of a network of extended families; this behavior was particularly useful in improving the somatic symptoms associated with depression in Japanese immigrants who identified with traditional culture [17].

Collectivism is very present in Russia and contrasts with the mostly individualistic conception of European countries. The characteristics of Russian collectivism are different still from Asian ones [18], likely because they arise from the need to contrast the harsh climate, impervious geophysical contexts (mountains, swamps, dense forests, etc.), hostile incursions, and history of serfdom [19]. Social support is enforced as unsolicited encouragement, care, and information [20], as well as unsolicited advice on health and practical matters [21]. The approach to the person is pragmatic and minimizes interpersonal harmony and autonomy [22]. In contrast to collectivism, however, a network of corrupt and individualistic connections described as a phenomenon called blat is used to obtain goods and services beyond formal procedures [16]. It is also interesting to note that Russians exert less control over their negative emotional expression with strangers, but more control over people they know, compared to American, Japanese, and Korean groups [19].

The dominant lifestyle in African cultures is collective, although a part of the population prefers individualized lifestyles; it is difficult to obtain objective assessments when quantitative measures are used, and specific comparisons are often required [23] since there is a tendency towards compliance [24]. However, it has also been found in these populations that pathologies such as depression are often expressed in somatic terms [13].

Given our broad research question and the heterogeneous body of literature and evidence currently available, we opted for the scoping review methodology. The main objectives investigated were related to the concept of acculturation of cultures in various communities and the concept of somatization as a culturally characterizing element. 

Regarding the concept of acculturation and culture, the purpose of this study was not to be limited exclusively to the geographical borders of a nation but to analyze the acculturation of culturally divergent groups of people (including the deaf community, Asian American women with breast cancer, immigrants from the Chernobyl area, Ghanaian teachers, intragenerational social mobility, etc.). The extent of somatization was then assessed based on the acculturation process according to the conclusions reported in the various papers.

## 2. Materials and Methods

The primary emphasis of a scoping review is on the breadth and relevance of the literature, thus including both qualitative and quantitative evidence. The quality of evidence in the included studies is of secondary concern and often difficult to operationalize due to the inclusion of such wide-ranging literature in scoping reviews.

The present review employed the scoping review methodology outlined by Arksey and O’Malley [25], with five stages: (1) developing a research question, (2) searching literature by using inclusion and exclusion criteria, (3) selecting articles, (4) charting data extracted from included articles, and (5) collating, summarizing, and reporting the findings. 

Articles matching the search criteria (somatization, medically unexplained symptoms, or functional disease and acculturation) were retrieved on PubMed until July 2021.

A total of 269 articles were identified. A first selection was made, and four articles with an irrelevant title, nine articles without abstracts, two duplicates, and four articles written in a language other than English were excluded. The abstracts of the remaining 250 articles were analyzed. Forty-seven articles were excluded because the abstracts were not relevant. The remaining 203 articles were then analyzed, and 43 articles matching the search criteria were identified [2,3,4,5,6,7,8,10,11,12,13,14,17,19,23,26,27,28,29,30,31,32,33,34,35,36,37,38,39,40,41,42,43,44,45,46,47,48,49,50,51,52].

## 3. Results

Chinese migrant populations were the most analyzed [3,4,7,12,14,27,30,31,45,46,53], followed by Latin populations [6,10,26,29,34,37,41,48,49,50]. The scales used to assess somatization and acculturation are different, with just a few studies having used the same, but all are self-assessment questionnaires to provide a standardized measure of an individual’s current psychological and/or psychopathological status.

The scales for assessing somatization differed. The most used was the Symptom Checklist-90 Revised (SCL-90R) (11 items), followed by the Brief Symptom Inventory (4 items). Other specific scales used were the Behavioural Assessment System for Children-2nd Edition [4], Hamilton Depression Scale (HAM-D) [32], Hamilton Anxiety Scale (HAM-A) [32,49], Brief Psychiatric Rating Scale (BPRS) [32], Semi-Structured Korean Interview Guide for Depression [11], PERI Demoralization Scale and Social Support Network Inventory [13], Hopkins Symptoms Checklist-25, Harvard Trauma Questionnaire [43], Impact of Event Scale [43], Memorial Symptom Assessment Scale [45,51], Harvard Trauma Questionnaire-Part I [43], PHQ-14 (PHQ somatic symptom scale) [48]. 

Table 1 shows some somatic aspects described. However, no study has evaluated the concomitant presence of syndromes directly related to the mind (e.g., irritable bowel syndrome, chronic fatigue syndrome), and only one study has evaluated pathologies that are affected by it, such as asthma [51]. Figure 1 shows how all the factors interact together causing a particular type of stress called acculturative stress.

Acculturation was assessed on different scales as well. The most used was the Suinn Lew Acculturation Scale [12,23,44]. Other scales were Acculturation Scale (Parker) [3], Acculturation Rating Scale for Mexican Americans [7], Behavioural Acculturation Scale [7], Asian American Family Conflict Scale [30], Brief Perceived Ethnic Discrimination Scale-Community Version (PEDQ-CV) [2], Everyday Discrimination Scale [33], Abbreviated Multidimensional Acculturation Scale (AMAS) [4], and Brief Acculturation Rating Scale for Mexican Americans-II (Spanish Version) [34]. 

## 4. Discussion

Somatization means transferring a psychological discomfort into somatic symptoms, which imply a disease that needs to be treated [54].

Immigration is a social, political, and economic event that leads people to leave their own country voluntarily or be forced to another land with different customs and cultures.

These people have to adapt to a new life and new habits. This is a process that can lead to clashes of culture and identity. In fact, adapting to a new culture often means modifying and transforming one’s identity and personality [55].

Immigrants want or need to create their own home in a new place, with people usually of different habits and culture. This period can be indefinite or limited in time for study or work reasons (Peace Corps, international students, embassy staff, etc.). Refugees and asylum seekers are people who have been violently forced to uproot themselves from their countries of origin for various reasons: wars, political problems, racial or gender laws, religious issues. 

Immigrants are people who have made a relatively free choice to relocate from one country, region or area to another. Theirs is a permanent decision to make their home in a new place. Sojourners are people who make a sustained but time-limited visit to live in another country. For example, Peace Corps workers, international students, and embassy staff can be considered sojourners. More recently, those entering countries as ‘guest workers’ on specific limited-time visa categories can also be considered in this category. Refugees and asylum seekers are people who have been forced to move from their home countries for various reasons.

The acculturation process in migrants and in future generations may be affected by mechanisms already modeled and tested in the cultural evolution literature, which could eliminate cultural variation between groups [56,57,58,59,60,61,62]. 

Immigration often leads a population with its cultural identity, ethnicity, habits, and customs to move to a territory with a different way of life. This can lead to a slow or fast change in the migrant population, which is reflected in the way of life, habits, and new laws and customs. Although genetic changes occur very slowly, there are still changes in psycho-physical conditions. In fact, cultural traits can change within a few generations. This is the process that tends to maintain cultural variation between groups, even if they undergo frequent migrations [63,64].

Somatization is expressed by individuals inconsistently and subjectively and is affected by the influence of various factors. Among these, a non-negligible aspect is attributed to the cultural context [30]; according to some interpretations, cultural groups systematically differ in their tendency to somatize instead of psychologizing their emotional distress [29,39].

The acculturation process is usually a major source of stress. Economic difficulties, discrimination, loss of social, family and support networks, and language gaps significantly affect mental health and psychological well-being, family and social relationships, development processes and more [65,66,67].

The phenomenon of migration leads to comparing one’s wealth of knowledge and visions of the world with different realities. Through this acculturative process, which is particularly stressful for the migrant and of variable duration and intensity, in favorable cases, a new balance is achieved, associated with good integration and an improvement in somatization concerning the extent of acculturation [36]. Different models of acculturation have been described where both the host culture and culture of origin can play principal roles. From this perspective, the concept of biculturalism is defined by an individual acquiring the host country’s culture while preserving the culture of their country of origin. Interestingly, biculturalism in specific groups has been associated with reduced somatization [33]. However, although the bicultural approach is associated with better integration, the process is quite complex [68,69]. Migrations are often heterogeneous phenomena and can include different dynamics that do not allow univocal conclusions to be drawn. The trauma suffered during migration, for example, could increase the incidence of somatization. For this reason, much attention has been given to the mental health of refugees because past experiences of war and violence, which add to the stressors of migration, could make them more vulnerable to PTSD (post-traumatic stress disorder), chronic pain, and somatic symptoms [70,71]. However, protective factors have also emerged: for those with strong religious beliefs, for example, a buffer role has been found when exposed to traumas of the migratory process of refugees [2].

The scientific evidence emphasizes the effects of migratory experiences on the mental health of migrants. In fact, specific stressors produce symptoms of anguish, anxiety, and depression through their cumulative effect. The traumatic events suffered during the whole migration process (before, during and after) are the main culprits, in addition to the change in place [72].

There is evidence that the risk of developing post-traumatic stress disorder (PTSD) is increased in clandestine migrants with pre-migration poverty [73]. This risk has also increased in poor neighborhoods and in post-migration discrimination [55,74]. Protective factors include a positive family environment and social support. Mental health effects in resettled refugee children in high-income countries were documented in a review in association with exposure to trauma, abusive parents, loss of one or more parents, host country discrimination, neighborhood violence, school isolation, and loss of one or more parents. On the other hand, the psychological well-being of the host environment, social support, stable settlement and religious beliefs were shown to be protective factors [75]. 

Family relationships are important for the development of a person during growth. In this context, the acculturation of parents shows direct correlations with the somatization of their children [76]. In fact, during the migratory and acculturation process, migrants must learn to recreate a new balance with a new social identity and self-image, often in difficult contexts. They have to find compromises between their new and old culture and learn to use new and different systems, sometimes without any support [77]. This mode prompts most parents to become bicultural. In this way, after evaluating which cognitions or practices of their own culture are more useful, they will modify the less suitable ones by choosing to acquire those of the new culture through experiences of deconstruction and reorganization. These individuals, if they reach a good balance, will manage to live harmoniously in certain situations with the culture of origin and the host culture [78,79,80,81].

Another interesting characteristic that has emerged from various works is that somatization in the migrant not only represents the expression of a transitory phase relating to acculturative stress but is also associated with psychological change. This is because the experiences lived by the migrant induce an adaptation in the individual that affects his way of somatizing [23]. Several studies emphasize that greater acculturation is associated with levels of somatization comparable to those of the host country [10,44]. This consideration is particularly evident in the studies with comparative group interviews conducted in both the migrant’s language of origin and in the host country’s language, which is indicative of acculturation [82]. These considerations underline the fact that the acculturation process induces changes of a psychological nature about somatic symptoms. The acculturation process, however, is complex, and it is not always possible to fully understand the dynamics. Indeed, conflicting studies have also been noted [7,12,83]. 

Social stress can induce somatic symptoms; this is possible even if an identifiable psychiatric disorder is not diagnosed [29,43,47,84]. Managing these symptoms is not always easy and can, in turn, create other problems such as emotional suppression or inhibition that can lead to further somatic discomfort. Emphasizing these symptoms, however, could distract attention away from recognizing social or interpersonal problems, thereby creating an additional psychosocial disadvantage [11]. 

Depressive symptoms in Latinas in the United States attributable to depression are not easily diagnosed, as cultural factors influence how they are presented [85,86]. In fact, their discomfort is often expressed in somatic symptoms through physical pain [87,88]. Since it is not always possible to identify them from a medical point of view, health resources are often used unnecessarily. Somatic symptoms in general are frequently associated with mental health problems. In the migrant, they can derive from stress factors that many Latinas may encounter. In Latinas, they can result from stressors that they may encounter. If untreated, persistent somatic symptoms can negatively affect physical and mental health [89]. 

Stress has also been recognized as an important risk factor for health-related conditions [90,91,92]. It is widely documented that psychosocial stressors are related to the development of organic diseases [93,94]; in particular, the stress experienced in childhood can influence the risks of various chronic physical diseases [95].

## 5. Conclusions

The migration process induces changes in the psyche that affect the nature of somatization. This change is also related to the acculturation process. The evaluation of somatization is measured according to non-specific symptoms, although several pathologies are closely related. The extent of symptoms and the subjective and objective expressions of diseases are now quantified in an increasingly reliable and specific manner by clinometry. Creating an updated rating scale that includes psychosomatic pathologies following the most recent clinimetric assessments would allow for a better understanding of the influence of socio-cultural contexts on the psyche in the genesis and management of mental-related pathologies, such as in the criteria for psychosomatic research. In this way, it would be possible to deploy treatment strategies more sensitive to specific cultures. There is evidence of how adapting the type of intervention to the cultural model can increase pro-social behavior and reduce children’s anxiety, hyperactivity, and somatization. 

## Figures and Tables

**Figure 1 ijerph-19-16011-f001:**
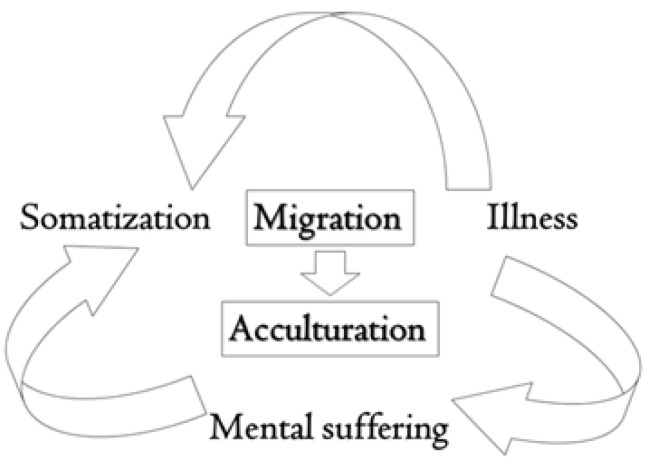
This diagram highlights the influence and interdependence of migration and acculturation on somatization. In migrants, a particular type of stress called acculturative stress accompanies the migratory process and manifests itself with high levels of somatization.

**Table 1 ijerph-19-16011-t001:** Characteristics of populations cited in the studies are included.

Population	Migration	Acculturation	Symptoms
Chinese	USA	Yes (knowledge of English, medium-high social status)	less pain sensation in patients with cancer
Chinese	USA	No (no knowledge of English)	increased depression and anxiety in breast cancer survivors
Chinese	USA	Yes and No	- decrease in chronic diseases such as diabetes, hypertension, arthritis. - increased depression (loss of appetite, fatigue, insomnia, sadness) linked to painful joint pathologies
Chinese	Australia	Yes and No	increased depression (loss of appetite, fatigue, insomnia, sadness) linked to painful joint pathologies
Chinese	USA	Yes and No	increase in neck and back pain
Hispanics	USA	No	increase in Seasonal Affective Disorder (SAD)
Hispanics	USA	No (no knowledge of English)	Increase in psychic depression (sadness, anhedonia, suicidal ideation, fatigue, lack of appetite, asthenia)
Hispanics	USA	No	Vasomotor symptoms, vaginal dryness, insomnia (in menopausal women)
Puerto Ricans	USA	Yes (intended as an integration of US culture)	decreased bronchial asthma, abdominal pain and headache
Koreans	USA	Yes and No	increase in gastric disturbances
Japanese	USA	Yes	Prevalent depressive symptoms were significantly lower
Soviet Jews	Israel	Yes and No	Increase in psychic depression (sadness, anhedonia, suicidal ideation, fatigue, lack of appetite, asthenia)
Soviet Jews from Chernobyl	USA	Yes and No	increase in chronic heart disease (e.g., arterial hypertension)
Turks	Belgium	Yes and No	Increased anxiety, sadness
Turks	Germany	Yes and No	Increase in cases of hyperemesis gravidarum
Ganese	No	Yes (increased schooling)	Increased depression and schizophrenia
Tanzanians	No	Yes (increased schooling)	Increased ERSAE-Stress-Prosocial (ESPS)
Swedes	No	Yes (increase in social position)	Reduction in joint and cardiovascular disorders
Somalis	Finland	Yes and No	Increased PTSD symptoms
Burundians	No	No	Increased PTSD symptoms

## Data Availability

Available upon request from the corresponding author.

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
