# Peer review of "Do Migration and Acculturation Impact Somatization? A Scoping Review"

_ijerph, 2022, doi:10.3390/ijerph192316011_

Round 1

Reviewer 1 Report

This paper reviews whether migration and acculturation impact somatization. It is of great significance.

However, there are still some problems in the article, and I hope that it can be revised accordingly.

1. Abstract: The abstract did not describe the research content enough. It needs to summarize the core conclusion of the paper. In the abstract section, which should summarize the article's content, the abstract should cover the background, the methods used, and the conclusions of the article.

2. Introduction: Although this article is a review, the introduction section should add appropriate context to the topic. For example, the background of migration and acculturation should be introduced.

3. Method: Need more details on the method. For example, why this method was chosen and the advantages of this method.

4. References: The majority of References are published in old days, Add more references in the new days. Particularly, papers from 2017 onwards were chosen more as references.

Author Response

Dear Reviewer,

we have modified the paper according to your requests point by point. The changes have been highlighted in red. Thank you for your suggestions that made it possible to clarify some passages of the manuscript that were not clear.

Reviewer: This paper reviews whether migration and acculturation impact somatization. It is of great significance.

However, there are still some problems in the article, and I hope that it can be revised accordingly.

1. Abstract: The abstract did not describe the research content enough. It needs to summarize the core conclusion of the paper. In the abstract section, which should summarize the article's content, the abstract should cover the background, the methods used, and the conclusions of the article.

Author: Dear Reviewer, thank you for the tips.

we have modified the abstract highlighting the main theme of the article: the role not only of the psyche in the onset of somatic pathologies, but also of acculturation in modifying the course

Reviewer: 2. Introduction: Although this article is a review, the introduction section should add appropriate context to the topic. For example, the background of migration and acculturation should be introduced.

Author: Dear Reviewer, thank you for the opportunity to improve our paper.

We have added what you have suggested us to underline, to better explain how migration is a touching moment for the migrant who leaves his homeland with a cultural and emotional baggage that influences his life.

Reviewer: 3. Method: Need more details on the method. For example, why this method was chosen and the advantages of this method.

Author: Dear Reviewer, thank you for the opportunity to explain us better.

As we stated at the end of the introduction section, “Given our broad research question and the heterogeneous body of literature and evidence currently available, we opted for the scoping review methodology.” According to Tricco et al., the three most common reasons for conducting a scoping review were to explore the breadth or extent of the literature, map and summarize the evidence, and inform future research. [Tricco, A. C., Lillie, E., Zarin, W., O’Brien, K., Colquhoun, H., Kastner, M., ... & Straus, S. E. (2016). A scoping review on the conduct and reporting of scoping reviews. BMC medical research methodology, 16(1), 1-10.]

In fact, Scoping reviews aim to provide an overview or map of evidence and are useful for examining emerging proof when it is still unclear what other, more specific questions can be posed for evidence syntheses and valuably addressed in adjunct it can be used to provide a broad overview of a topic.

Therefore, we modified the section according your suggestion.

The primary focus emphasis of a this scoping review is on the breadth and relevance of literature, thus including both qualitative and quantitative evidence to examine the direct link between migration and acculturation on somatization. The quality of evidence in the included studies is of secondary concern and often difficult to operationalize due to the inclusion of such wide-ranging literature in scoping reviews.

The methodology of present review employed the scoping-review is based on the framework methodology outlined by Arksey and O’Malley, with five stages: (1) developing identifying a research question, (2) Identifying relevant studies searching literature by using inclusion and exclusion criteria, (3) studies selection selecting articles, (4) charting data extracted from included articles, and (5) collating, summarizing, and reporting the findings.

Articles matching the search criteria (somatization, medically unexplained symptoms, or functional disease and acculturation) were searched on PubMed until July 2021.

A total of 269 articles were identified. A first selection was made, and four articles with an irrelevant title, nine articles without abstracts, two duplicates, and four articles written in a language other than English were excluded.

The abstracts of the remaining 250 articles were analysed. We excluded articles that did not have full text available, and ones that did not refer to specific somatic symptoms or were not associated with the migration process. Forty-seven articles were excluded because the abstracts were not relevant. The remaining 203 articles were then analysed, and 43 articles matching the search criteria were identified [2,3,13,14,17,19,23,26–30,4,31–40,5,41–50,6,51,52,7,8,10–12].”

Reviewer: 4. References: The majority of References are published in old days, Add more references in the new days. Particularly, papers from 2017 onwards were chosen more as references.

Author: Dear Reviewer, thank you for consideration. We have updated the literature according to your indications. The topic we are dealing with has been dealt with more exhaustively in older articles. However many authors have provided recent interpretations of the theme which we have updated

Reviewer 2 Report

Barbati and colleagues wrote a paper entitled "Do migration and acculturation impact somatization?" a scoping review. ". The manuscript aimed to show the psychological problems that different ethnic groups go through during migration, which lead to somatization. As migrations are increasingly present in the modern world and represent a global problem, I believe that it is important to influence the consciousness of the broader community through the system of science in order to enable these groups of the world's population to integrate as well as possible.

This topic is interesting.

Conceptually, the manuscript is well laid out, although it could be more clearly and fluently written.

It is necessary to correct the following:

1. The paper uses old literature; only 16% of the literature used is within the last five years, which I consider inadequate. Given that it is a review paper with such a current topic, at least 30% of the cited literature should be within the last five years.

2. The literature needs to be cited adequately (MDPI). In some references, for example (13; 15), the year of publication of the articles needs to be specified.

3. By checking for plagiarism in the Turnitin program, a match with other articles was observed in:

Lines 103-110

Lines 155-167

Author Response

Manuscript ijerph-2036152

Title: “Do migration and acculturation impact somatization? a scoping review.

Dear Editor in Chief Prof. Dr. Paul B. Tchounwou,

thank you for giving us the opportunity to change and improve our paper titled “Do migration and acculturation impact somatization? a scoping review”. Below we have reported our answers to the Reviewers' comments point by point. Furthermore, the revision also considered the Editorial suggestions. All authors shared the changes reported in the paper

Best Regards

The corresponding author

Antonello Barbati

Reviewer 2

Dear Reviewer,

we have modified the paper according to your requests point by point. The changes have been highlighted in red. Thank you for your suggestions that made it possible to clarify some passages of the manuscript that were not clear.

Reviewer: Barbati and colleagues wrote a paper entitled "Do migration and acculturation impact somatization?" a scoping review. ". The manuscript aimed to show the psychological problems that different ethnic groups go through during migration, which lead to somatization. As migrations are increasingly present in the modern world and represent a global problem, I believe that it is important to influence the consciousness of the broader community through the system of science in order to enable these groups of the world's population to integrate as well as possible.

This topic is interesting.

Conceptually, the manuscript is well laid out, although it could be more clearly and fluently written.

It is necessary to correct the following:

1. The paper uses old literature; only 16% of the literature used is within the last five years, which I consider inadequate. Given that it is a review paper with such a current topic, at least 30% of the cited literature should be within the last five years.

Author: Dear Reviewer thank you for your suggestion. We have updated the literature according to your indications. The topic we are dealing with has been dealt with more exhaustively in older articles. However many authors have provided recent interpretations of the theme which we have updated

Reviewer: 2. The literature needs to be cited adequately (MDPI). In some references, for example (13; 15), the year of publication of the articles needs to be specified.

Author: Dear Reviewer thank you for your observation, we added it.

Reviewer: 3. By checking for plagiarism in the Turnitin program, a match with other articles was observed in: Lines 103-110; Lines 155-167.

Author: Dear Reviewer thank you for the opportunity to explain us better; we modified it.

Round 2

Reviewer 2 Report

Barbati and co-autors partially accepted the recommended changes.

It is necessary to correct the following:

1.       The literature needs to be cited adequately (MDPI).

2.       By checking for plagiarism in the Turnitin program, a match with other articles was observed in Discussion: Lines1-8; Lines10-13; Lines 15-17.

Author Response

Reviewer 2

Dear Reviewer,

we have modified the paper according to your requests point by point. The changes have been highlighted in red. Thank you for your suggestions that made it possible to clarify some passages of the manuscript that were not clear.

Reviewer: Barbati and co-autors partially accepted the recommended changes. It is necessary to correct the following: The literature needs to be cited adequately (MDPI).

Author: Dear reviewer, thank you for the observation. Maybe it was a typo in assembling the paper with all the corrections. We have now submitted the correct paper.

Reviewer: By checking for plagiarism in the Turnitin program, a match with other articles was observed in Discussion: Lines1-8; Lines10-13; Lines 15-17.

Author: Dear Reviewer, thank you for the opportunity to improve our paper. We modify it:

4. Discussion

Somatization means transferring a psychological discomfort into somatic symptoms, which imply a disease that needs to be treated [54].

Immigration is a social, political and economic event that leads people to leave their own country voluntarily or forced to another land with different customs and cultures.

These people have to adapt to a new life and new habits. This is a process that can lead to clashes of culture and identity. In fact, adapting to a new culture often means modifying and transforming one's identity and personality [55].

Immigrants want or need to create their own new home in a new place, with new people usually of different habits and culture. This period can be indefinite or limited in time for study or work reasons (Peace Corps, international students, embassy staff, etc...). Refugees and asylum seekers are people who have been violently forced to uproot themselves from their countries of origin for various reasons: wars, political problems, racial or gender laws, religious issues.

Somatization implies a tendency to experience and communicate psychological distress in the form of somatic symptoms and to seek medical help for them [54].

Immigration is a central process that contributes to the cultural diversification of nation-states. Historically, groups of people have moved voluntarily or displaced, often losing a sense of belonging. These people are usually required to adapt to other groups' social settings and norms. Central to the displacement and the subsequent transition and settlement is the threat to culture and identity. Displacement and dislocation usually mean a disruption of taken-for-granted social and support systems, cultural rules, rituals, symbols, and meanings central to identity construction and personhood [55].